# Effects of childhood trauma on mental health outcomes, suicide risk factors and stress appraisals in adulthood

**Leizhi Wang**[ID]*, **Chris Keyworth**[ID], **Daryl B. O'Connor**

School of Psychology, University of Leeds, Leeds, United of Kingdom

* ps22lw2@leeds.ac.uk

## Abstract

### Background

Childhood trauma has been identified as a significant risk factor for adverse mental health outcomes and increased suicide risk. However, the mechanisms by which stress-related variables, such as stress appraisal, influence this pathway remain unclear. The current study sought to: (1) investigate whether childhood trauma was associated with mental health outcomes (depression, anxiety), suicide risk factors (defeat, entrapment), and stress related outcomes (stress appraisal, perceived stress) in adulthood and (2) determine whether stress appraisal and perceived stress mediate the effects of childhood trauma on mental health outcomes and suicide risk factors.

### Method

273 participants were recruited to an online prospective study that consisted of two sessions with a one-week interval. In session 1, the childhood trauma questionnaire was completed. In session 2, participants completed measures assessing the level of depression, anxiety, defeat, entrapment, stress appraisal and perceived stress over the preceding week.

### Results

Analyses found that childhood trauma was significantly associated with higher scores on depression, anxiety, defeat, entrapment, stress appraisal and perceived stress. Childhood trauma also had indirect effects on depression, anxiety, defeat, and entrapment through stress appraisal and perceived stress.

### Conclusion

These findings underscore the enduring impact of childhood trauma on mental health outcomes and suicide risk in adulthood, mediated through its influence on stress

**Data availability statement:** The anonymised dataset underpinning this study has been deposited in the Figshare public repository and is freely accessible (https://doi.org/10.6084/m9.figshare.29088083.v1).

**Funding:** The author(s) received no specific funding for this work.

**Competing interests:** The authors have declared that no competing interests exist.

appraisals and perceptions of stress encountered in daily life. These underlying mechanisms are critical for informing the development of future interventions.

## Introduction

Childhood trauma is experienced by nearly one third of young people in the UK [1]. The impact of childhood trauma on its victims is both profound and enduring [2]. Exposure to childhood trauma has been consistently associated with an elevated risk of developing depression and anxiety in adulthood. Felitti et al. [3] revealed a strong, positive relationship between the number of adverse experiences in childhood and the likelihood of developing depression and anxiety in later life. Similarly, previous research has also found that childhood trauma is significantly associated with negative mental health outcomes including depression, anxiety, posttraumatic stress disorder (PTSD), and substance abuse [4,5]. Furthermore, childhood trauma has been extensively linked to an increased risk of suicidal ideation and suicide attempts in later life. Dube et al. [6] found that individuals who experienced multiple forms of childhood trauma were significantly more likely to exhibit suicidal behaviours. A meta-analysis by Devries et al. [7] demonstrated that females who experienced childhood sexual abuse had a substantially higher likelihood of attempting suicide compared to those without such experiences.

Recent research has focused on exploring the links between childhood trauma and two critical risk factors for suicide: defeat and entrapment. Previous studies have demonstrated that higher levels of childhood trauma are also linked to increased feelings of defeat and entrapment [8]. In another study employing a seven-day daily diary method, childhood trauma was found to be positively correlated with a range of suicide risk indicators, including diminished reasons for living, decreased optimism, daily thoughts of suicide, and heightened perceptions of defeat and entrapment [9]. Defeat refers to a sense of failed struggle and loss of social status or identity, while entrapment is the perception of having no escape from these adverse circumstances [10]. According to the Integrated Motivational-Volitional (IMV) model of suicidal behaviours [11,12], perception of defeat and entrapment are important factors in the formation of suicidal ideation. The IMV model offers a comprehensive framework for understanding the complex processes leading to suicide, dividing its development into three stages: pre-motivational, motivational, and volitional. It states that the transition that occurs during the motivational stage is facilitated by a number of factors, among which defeat, and entrapment are two most pivotal risk factors.

Although the effects of childhood trauma on mental and physical health outcomes have been increasingly investigated, significant gaps remain in our understanding of the mechanisms linking childhood trauma to mental health outcomes and proxies for suicidal risk, in particular, the role played by stress-related variables. Therefore, one of the aims of the current study is to investigate the role of perceived stress and stress appraisals in the effects of childhood trauma in adulthood. Stress has long been recognised as a critical pathway through which childhood trauma affects subsequent health outcomes. Anda et al. [13] and Shonkoff et al. [14] emphasised the

long-term impacts of acute stress events in early childhood on the development of the physiological stress system and the nervous system. These prolonged effects on the stress system lead to a range of negative health consequences later in life, such as increasing mental health concerns [2]. However, no work has explored the effect of childhood trauma on how individuals appraise daily stressors.

Daily hassles, or daily stressors, refer to the day-to-day events that produce negative feelings such as annoyance, irritation, and worry or frustration [15]. Early studies have demonstrated that the cumulative effects of daily hassles on health may surpass the impact of acutely stressful events [16,17]. Kanner et al. [18] posited that conventional indices of acute stress fail to capture the nuances of everyday life, asserting that it is the everyday events that bear direct relevance to health outcomes.

Recent studies have demonstrated that childhood trauma is associated with increased levels of perceived daily stress or hassles [8,9]. In a study by O'Connor et al. [19], daily stress was found to mediate the relationship between childhood trauma and various sleep indicators, including sleep quality, total sleep duration, and sleep onset latency. However, measurements of perceived stress do not account for individuals' subsequent cognitive processing of stressors, such as appraisal; rather, they reflect only the perception of the event's stressfulness. Cognitive appraisal of a stressor refers to how an individual evaluates whether a noxious event is threatening, harmful or loss-provoking, and is a central component of the transactional model of stress [20]. The transactional model posits that the appraisal of a stressful event can be divided into two corresponding stages: primary and secondary appraisal. In terms of daily hassles appraisal, in the primary appraisal, an individual assesses how much these hassles matter to them, which focus on the evaluation of the event itself. In the secondary appraisal, an individual evaluates the available coping strategies to change the outcome or make things better [21]. An individual will only experience stress when they perceive a mismatch between primary and secondary appraisals, such that there is a discrepancy between perceived demands and available resources [22]. Surprisingly, no research has investigated whether childhood trauma influences the appraisal of stressful events in adulthood or perceptions of stress more generally or whether stress appraisals may mediate the effects of childhood trauma on mental health outcomes. Therefore, one of the aims of the current study was to investigate whether childhood trauma had indirect effects on these outcomes through influencing perceived stress and stress appraisals.

The effects of childhood trauma on mental and physical health outcomes in adulthood has also been found to be influenced by social support and socioeconomic status (SES) [23,24]. Previous research [25] has found that the effect of childhood trauma on depression and anxiety was significantly moderated by social support which served as the protective factors. The IMV model [12] also pointed out that the social support is an important motivational moderator which can influence the transition from the stage of entrapment to the formation of suicidal ideation and intent.

In addition to social support, SES is also regarded as a significant moderator in the relationship between childhood trauma and mental health outcomes. Lower SES exacerbates the negative effects of childhood trauma on sleep quality and other stress-related indicators [26]. Notably, previous studies have measured objective SES, which represents individuals' objective income and education level. In contrast, subjective SES is based on an individual's personal perception of their socioeconomic position or rank within society [27]. Adler et al. [28] found a positive association between subjective SES and cortisol habituation to repeated stress, and this association remained significant even after the objective SES was controlled. Moreover, subjective SES is sensitive to contextual variations and cultural differences in perceptions of social hierarchy, making it a more versatile tool in diverse populations [29].

Taken together, the primary aims of the current study were three-fold:

1. To investigate the effects of childhood trauma on stress-related outcomes (perceived stress, stress appraisals), mental health outcomes (depression, anxiety), and suicide risk factors (defeat, entrapment).

2. To determine whether stress-related variables (perceived stress and stress appraisals) mediate the effects of childhood trauma on the mental health outcomes and suicide-risk factors.

3. To test whether social support, subjective socioeconomic status, and past suicide-related history moderate any of the associations between childhood trauma and three group of outcomes (mental health, stress-related, and suicide risk factors).

## Method

### Participants and study design

This study adopted an online prospective design with data collection across two sessions. In Session 1, participants completed a series of questionnaires that assessed their demographic background, history of childhood trauma, perceived social support, subjective socioeconomic status, and suicide-related experiences. One week after completing Session 1, participants were required to complete the Session 2 survey which included another set of questionnaires that measured daily stress appraisals, perceived stress, depression severity, anxiety severity, perceived defeat, and perceived entrapment. All participants were recruited from the Prolific Academic online data collection platform. Participants were included if they were over 18 years old and were native English speakers. Participants were excluded if they failed to answer the attention check questions correctly (N = 53). Upon completing the two sessions, participants received a total cash reward of £3 (£1.5 per session).

The data collection for this study commenced on August 10, 2024, and concluded on August 23, 2024. Written informed consent was obtained from all participants, and their data were anonymized, ensuring the absence of identifiable information.

The current study was approved by the Research Ethics Committee of the School of Psychology, University of Leeds (REC reference: PSCETHS-1110), and the main study hypotheses were preregistered at AsPredicted.org (#182080).

### Sample size estimation

Since this study is, to our knowledge, the first to investigate the effects of childhood trauma on stress appraisals—particularly in relation to daily stress—the existing literature lacks directly relevant parameters. Therefore, to achieve the optimal power, the sample size of current study was estimated by a power analysis using G*Power which was based on 7 previous published studies that either reported the correlational relationship between childhood trauma and perceived stress, or childhood trauma and daily hassles [30–36]. The average correlation coefficient across these seven studies was r = 0.20. The power analysis showed that the minimum sample size needed was N = 266 for the bivariate correlation model (α = .05; power = .95). In order to allow for 15% attrition and drop out between two sessions, the current study aim to recruit 300 participants.

### Questionnaire measures

*History of childhood trauma* was measured using the Childhood Trauma Questionnaire (CTQ) [37]. The CTQ is a 28-item self-report measure which was designed to assess retrospective reports of childhood abuse and neglect experiences. The CTQ contains five sub-scales for five different types of childhood trauma: emotional abuse, physical abuse, sexual abuse, emotional neglect, and physical neglect. Respondents are asked to rate each item on a scale ranging from 1 = never true to 5 = very often true. The CTQ has been shown to have good reliability and validity [37] and in the current sample it had good internal consistency (α = 0.94).

*Daily stress appraisals* were measured using the Daily Stressor Appraisal Scale (DSAS) [21]. The DSAS is an 8-item self-reported questionnaire (5 items assessed primary appraisals, 3 items assessed secondary appraisals). Participants are asked to think of the most stressful daily hassle they have experienced in the last 7 days either at work or at home and to respond to each item on a scale from 1 (not at all) to 7 (to a very large extent). A total stress appraisal score was calculated by dividing the primary appraisal mean score by the secondary appraisal mean score. A higher stress appraisal

score (e.g., higher ratio between two appraisals) indicates an imbalance between perceived demands and perceived resources to meet the demands. The DSAS has been shown to have good reliability and validity [21,38]. In the current sample, the DSAS demonstrated good internal consistency with α = 0.92 for the primary appraisal items, and α = 0.81 for the secondary appraisal items.

*Perceived stress* was measured using the Perceived Stress Scale – Short Form (PSS-Brief) [39]. The PSS-Brief is a self-report questionnaire comprising four items asking participants about the past 7 days. Participants responded on a four Likert scale from 0 = Never to 5 = Very Often. The PSS has been shown to have good reliability and validity [40]. In the current sample, the PSS displayed good internal consistency with α = 0.71.

*Depression symptoms* were assessed using the Patient Health Questionnaire (PHQ-9) [41]. This 9-item questionnaire assesses depression symptoms over the last two weeks on a four-point Likert scale from 0 = Not at all to 3 = Nearly every day. The total score ranges from 0–27. The PHQ-9 has been shown to have good reliability and validity [41]. The PHQ-9 demonstrated good reliability in the current sample (α = 0.90).

*Anxiety symptoms* were assessed using the Generalised Anxiety Disorder Assessment (GAD-7) [42]. Seven items are scored on a four-point Likert scale ranging from 0 = Not at all to 3 = Nearly every day. The total score ranges from 0–21. The GAD-7 has been shown to have good reliability and validity [43]. In current study, GAD-7 demonstrated a good internal consistency (α = 0.92).

*Defeat* was assessed using the 16-item Defeat Scale (D-16) [44]. Example items included "I feel completely knocked out of action" were rated on a five-point Likert scale ranging from 0 = Never to 4 = Always. Higher total scores of all items indicated greater levels of defeat. The defeat scale has been shown to have good reliability and validity [45]. In the current sample, the defeat scale displayed good internal consistency with α = 0.96.

*Entrapment* was assessed using the Entrapment Scale (E-16) [44]. Respondents are asked to rate each item on a scale ranging from 0 = Never to 4 = Always. The sum of the 16 items indicates the current levels of entrapment. The entrapment scale has been shown to have good reliability and validity [45] and in the current sample it has good internal consistency (α = 0.96).

*Social support* was measured using the Multidimensional Scale of Perceived Social Support (MSPSS) [46]. Responses are rated on a 7-point Likert scale ranging from 1 = "Very Strongly Disagree" to 7 = "Very Strongly Agree." Higher total scores indicate a greater level of perceived social support. MSPSS has been shown to have good reliability and validity [47] and also displayed good reliability in the current sample with Cronbach's α = 0.90.

*Subjective Socioeconomic Status* was measured using the MacArthur Scale of Subjective Social Status (MacArthur SSS Scale) [28]. Participants were shown an image of a ladder with 10 steps. They were informed that the ladder represents the social hierarchy, with the top step symbolising the highest level of socioeconomic status and the bottom step representing the lowest level. Participants were asked a single item: to indicate where they perceive themselves to be positioned on this ladder. Their responses were recorded on a 10-point Likert scale, reflecting their perceived subjective socioeconomic status.

*Lifetime suicide-related and self-harm behaviour* was measured using the Adult Psychiatric Morbidity Survey (APMS) [48]. Three items were taken from APMS to determine lifetime suicide ideation: "Have you ever seriously thought of taking your life, but not actually attempted to do so?", lifetime suicide attempt: "Have you ever made an attempt to take your life, by taking an overdose of tablets, or in some other way?", and lifetime self-harm behaviour: "Have you ever deliberately harmed yourself in any way but not with the intention of killing yourself?" Participants may select one response from three options: 'Yes', 'No', or 'Prefer not to say'.

## Statistical analysis

Missing values were replaced with the mean value of the respective item within the dataset. All analyses were performed using IBM SPSS Statistics version 28.0.1. A series of linear regression models were conducted to assess whether levels

 

of childhood trauma significantly predicted six outcomes across three groups (stress-related outcomes, suicide risk factors and mental health outcomes).

The assumptions of homoscedasticity and linearity were checked. If any regression model violated the homoscedasticity assumption, then, following the recommendations of Field [49], regression analyses employed a bootstrap approach with 5,000 resamples and utilized the bias-corrected and accelerated (BCa) method to estimate confidence intervals, thereby mitigating the effects of heteroscedasticity. Note that all linear regression analyses of the CTQ subscales were conducted, and all their results were statistically significant (all p < .01). Therefore, to be more parsimonious and reduce multiple comparisons, the CTQ subscales were not considered further in the current paper.

Next, a series of mediation models were conducted also using the PROCESS macro. The total CTQ score and the two groups of outcomes served as the independent variable (X) and the dependent variables (Y), respectively. The stress-related variables acted as mediators (M variables). 5000 bias corrected bootstrap samples were used to generate 95% confidence interval and point estimates for indirect effects.

Lastly, a series of moderation models were conducted using the PROCESS macro in SPSS [50]. Moderation models were conducted between CTQ total score and six outcome variables. Using the moderation analysis, the moderation effect of subjective SES, perceived social support and suicide-related history were explored in any observed direct effect. Johnson-Neyman Technique was implemented when the moderator was a continuous variable.

## Results

Two hundred and seventy-three participants had a mean age 38 (SD = 12.77), and 132 (48.4%) of them were male, with the largest response group was white (85%) and employed (78.4%). Descriptive statistics for all the study variables are presented in Table 1. We observed similar CTQ scores (M = 42.44, SD = 15.99) compared to previous research [9,19].

Table 2 presents the correlation results for all study variables. As shown, the total CTQ score was significantly positively correlated with the mental health outcomes and the suicide risk factors indicating that higher levels of trauma were associated with more adverse outcomes.

Regarding stress-related variables, the analysis showed significant positive correlations between the CTQ total score and both primary stressor appraisal (r (271) = 0.19, p = 0.002) and perceived stress level (r (271) = 0.35, p < 0.001) indicating higher childhood trauma is associated with higher perceptions of stress in adulthood. Meanwhile, the analysis also found a significant negative correlation between the CTQ total score and secondary stressor appraisal (r (271) = −0.21, p < 0.001),

**Table 1. Descriptive statistics for study variables.**

| Study Variable | M | SD |
|---|---|---|
| Childhood Trauma | 42.44 | 15.99 |
| Stress Appraisals | 1.61 | 1.01 |
| Perceived Stress | 6.81 | 3.22 |
| Depression | 7.60 | 6.11 |
| Anxiety | 6.98 | 5.36 |
| Defeat | 22.81 | 13.90 |
| Entrapment | 17.36 | 15.10 |
| Social Support | 64.02 | 12.97 |
| Subjective SES | 5.40 | 1.48 |

M, Mean; SD, Standard deviation; SES, socioeconomic status; N = 273.

Score ranges for study variables: childhood trauma (25–125), stress appraisals (0.1–7), perceived stress (0–16), depression (0–27), anxiety (0–21), defeat (0–64), entrapment (0–64), social support (12–84), subjective SES (1–10).

**Table 2.  Correlations between main study variables.**

| Variables | r | | | | | | | |
|---|---|---|---|---|---|---|---|---|
| | 1 | 2 | 3 | 4 | 5 | 6 | 7 | 8 |
| 1. **CTQ Total Score** | – | | | | | | | |
| 2. **Stress Appraisals** | .24** | – | | | | | | |
| 3. **Perceived Stress** | .35** | .64** | – | | | | | |
| 4. **Depression** | .38** | .55** | .72** | – | | | | |
| 5. **Anxiety** | .34** | .60** | .71** | .77** | – | | | |
| 6. **Defeat** | .39** | .64** | .76** | .81** | .73** | – | | |
| 7. **Entrapment** | .40** | .56** | .72** | .78** | .73** | .78** | – | |
| 8. **Social Support** | −.35** | −.21** | −.34** | −.33** | −.24** | −.41** | −.34** | – |
| 9. **Subjective SES** | −.22** | −.24** | −.26** | −.28** | −.20** | −.36** | −.22** | .17** |

*\*\*p<0.01. N=273*

indicating that more severe childhood trauma experiences were associated with poorer judgments of one's ability to cope. Consistent with these findings, the results also showed that the total CTQ score had a significant positive correlation with the stress appraisal score ($r$ (271) = 0.24, $p<0.001$), suggesting that more severe childhood trauma experiences correlated with greater stress appraisals—characterised by greater demands (primary appraisal) being matched with fewer resources (secondary appraisal).

### Direct effects of childhood trauma on stress related outcomes (perceived stress, stress appraisals), mental health outcomes (depression, anxiety), and suicide risk outcomes (defeat, entrapment)

A series of linear regression models were conducted to test whether the CTQ total score could significantly predict three groups of outcomes (mental health, stress-related factors, and suicide risk factors) measured one week later. In all regression analyses, age and gender were controlled. One participant who identified as a transgender male was categorised within the male group. Six participants who reported a gender identity other than male or female (4 as non-binary; 2 prefer not to say) were excluded from the current regression analysis. The final sample size for the regression analysis was 267.

Table 3 summarises all the results of the regression analyses. According to the findings, the total score of CTQ significantly predicted all outcomes across the three groups of variables. Specifically, the total CTQ score significantly predicted stress appraisals ($b=0.02$, 95% CI [0.01, 0.023], $p<0.001$), perceived stress ($b=0.07$, 95% CI [0.05, 0.10], $p<0.001$), depression ($b=0.15$, 95% CI [0.10, 0.20], $p<0.001$), anxiety ($b=0.12$, 95% CI [0.07, 0.16], $p<0.001$), defeat ($b=0.35$, 95% CI [0.23, 0.47], $p<0.001$), and entrapment ($b=0.39$, 95% CI [0.26, 0.51], $p<0.001$). These results illustrate that greater exposure to childhood trauma significantly predicted poorer mental health, stress appraisals, and indicators of suicide risk one-week later.

### Indirect effects of childhood trauma on depression, anxiety, defeat and entrapment via stress-related variables

**Depression and anxiety.**  A summary of the indirect effects is presented at Table 4. There were significant indirect effects of childhood trauma on depression symptoms through both stress-related variables: perceived stress ($b=0.09$, 95% CI [0.05, 0.12], $p<0.001$) and stress appraisals ($b=0.04$, 95% CI [0.02, 0.07], $p<0.001$). *Similarly,* there were significant indirect effects of childhood trauma on anxiety symptoms through both stress-related variables: perceived stress ($b=0.08$, 95% CI [0.05, 0.11], $p<0.001$) and stress appraisals ($b=0.04$, 95% CI [0.02, 0.07], $p<0.001$), which is shown in upper panel of Fig 1.

**Defeat and entrapment.**  There were significant indirect effects of childhood trauma on defeat through both stress-related variables: perceived stress ($b=0.21$, 95% CI [0.13, 0.30], $p<0.001$) and stress appraisals ($b=0.12$, 95% CI [0.06,

**Table 3. A summary of the main effects of childhood trauma on stress appraisals, perceived stress, depression, anxiety, defeat, and entrapment.**

| Variable | Step 1 | | | Step 2 | | |
|---|---|---|---|---|---|---|
| | *SE* | *Coefficient* | *p* | *SE* | *Coefficient* | *p* |
| **Stress Appraisals** | | | | | | |
| Intercept | 0.173 | 0.958 | <.001** | 0.270 | 1.222 | <.001** |
| CTQ | 0.004 | 0.016 | <.001** | 0.004 | 0.015 | .001* |
| Age | | | | 0.004 | −0.005 | .284 |
| Gender | | | | 0.119 | −0.149 | .216 |
| **Perceived Stress** | | | | | | |
| Intercept | 0.582 | 3.746 | <.001** | 0.799 | 5.337 | <.001** |
| CTQ | 0.014 | 0.072 | <.001** | 0.014 | 0.073 | <.001** |
| Age | | | | 0.014 | −0.043 | .002* |
| Gender | | | | 0.366 | 0.094 | .806 |
| **Depression** | | | | | | |
| Intercept | 1.032 | 1.236 | .232 | 1.467 | 4.788 | .002* |
| CTQ | 0.026 | 0.150 | <.001** | 0.025 | 0.150 | <.001** |
| Age | | | | 0.024 | −0.097 | <.001** |
| Gender | | | | 0.668 | 0.200 | .764 |
| **Anxiety** | | | | | | |
| Intercept | 0.889 | 2.065 | .024 | 1.328 | 5.539 | <.001** |
| CTQ | 0.020 | 0.115 | <.001** | 0.020 | 0.115 | <.001** |
| Age | | | | 0.022 | −0.086 | <.001** |
| Gender | | | | 0.593 | −0.389 | .518 |
| **Defeat** | | | | | | |
| Intercept | 2.509 | 8.134 | .002* | 3.344 | 15.430 | <.001** |
| CTQ | 0.061 | 0.345 | <.001** | 0.060 | 0.346 | <.001** |
| Age | | | | 0.053 | −0.198 | <.001** |
| Gender | | | | 1.553 | 0.354 | .824 |
| **Entrapment** | | | | | | |
| Intercept | 2.668 | 1.077 | .682 | 3.846 | 6.611 | .086 |
| CTQ | 0.066 | 0.381 | <.001** | 0.064 | 0.385 | <.001** |
| Age | | | | 0.062 | −0.185 | .003* |
| Gender | | | | 1.671 | 2.651 | .118 |

Note: N = 267. CTQ, childhood trauma questionnaire, SE = Coefficients Standard Error,

5000 bootstrap approaches with bias-corrected and accelerated (BCa) method to estimate confidence intervals.

*Indicates statistical significance at p < .01; **Indicates statistical significance at p < .001.

0.20], p < 0.001). Moreover, there were significant indirect effects of childhood trauma on entrapment through both stress-related variables: perceived stress (b = 0.21, 95% CI [0.13, 0.30], p < 0.001) and stress appraisals (b = 0.11, 95% CI [0.05, 0.18], p < 0.001), which is shown in lower panel of Fig 1.

## Moderating effects of social support, subjective SES, and suicide related history on the childhood trauma – three groups of outcomes relationship

A series of moderation analyses were conducted to test the interaction effects of the total CTQ score on three groups of outcomes via three moderators using PROCESS. All results of them were non-significant (see S1 Fig for the summary of the results).

**Table 4. Summary of all the indirect effects.**

| Indirect effects | b (unstandardised) | SE | 95% CI | p |
|---|---|---|---|---|
| **Outcome: Depression** | | | | |
| 1.1. CTQ – stress appraisals – depression | 0.043 | 0.012 | [0.020, 0.074] | <.001 |
| 1.2. CTQ – perceived stress – depression | 0.088 | 0.016 | [0.055, 0.123] | <.001 |
| **Outcome: Anxiety** | | | | |
| 2.1. CTQ – stress appraisals – anxiety | 0.044 | 0.012 | [0.020, 0.074] | <.001 |
| 2.2. CTQ – perceived stress – anxiety | 0.079 | 0.014 | [0.049, 0.111] | <.001 |
| **Outcome: Defeat** | | | | |
| 3.1. CTQ – stress appraisals – defeat | 0.120 | 0.040 | [0.058, 0.198] | <.001 |
| 3.2. CTQ – perceived stress – defeat | 0.213 | 0.037 | [0.131, 0.298] | <.001 |
| **Outcome: Entrapment** | | | | |
| 4.1. CTQ – stress appraisals – entrapment | 0.111 | 0.030 | [0.051, 0.184] | <.001 |
| 4.2. CTQ – perceived stress – entrapment | 0.215 | 0.038 | [0.134, 0.300] | <.001 |

5000 bootstrap, N = 273

## Discussion

Four key findings emerged from the current study. First, childhood trauma significantly influenced mental health outcomes, including higher depression and anxiety symptoms. Second, childhood trauma was also positively associated with higher defeat and entrapment which are important indicators for suicide risk. Third, childhood trauma was positively associated with stress appraisals and perceived stress. Lastly, the current study found that the effects of childhood trauma on mental health outcomes and suicide risk factors were mediated through stress appraisals and perceived stress.

It is well-established that childhood trauma has serious long term negative consequences which increase the levels of depression and anxiety symptoms. Findings from the current study are consistent with previous literature. An earlier study, comprising 4,141 structured interviews [51], found that approximately half of the patients with anxiety disorders reported experiences of childhood physical or sexual abuse. A meta-analysis by Gardner et al. [52] also reported that all-forms of childhood maltreatment were significantly associated with depression, while several types of childhood trauma were significantly associated with anxiety and PTSD.

The current study also found that childhood trauma was associated with higher defeat and entrapment in later adulthood. According to the IMV model [12], both defeat and entrapment are pivotal risk factors which transit the individual from pre-motivation of suicide to suicidal ideation. This finding is consistent with earlier work showing a strong association between childhood trauma and suicide risk factors in later life [53–57]. It is also worth noting that research in this area has been reliant on cross-sectional designs, whereas a growing body of work has utilised naturalistic longitudinal designs such as the daily diary approaches [8,9,19]. These latter approaches aim to address these shortcomings, therefore, future research should continue to adopt prospective and longitudinal designs.

The current study also found that childhood trauma experienced in early life is associated with higher perceived stress in adulthood. This is consistent with Rogerson et al. [9] who also found that childhood trauma was associated with greater daily perceived stress and poorer sleep quality. Similarly, O'Connor et al. [19] also found that childhood trauma was associated with higher daily perceived stress and perseverative cognition. Moreover, the current study is the first to show that childhood trauma is associated with higher stress appraisals. In other words, we found that individuals with a history of childhood trauma are more likely to experience stress because they perceive that they do not have the coping resources to deal with the demands of stressful daily hassles.

The current results also found that the effects of childhood trauma on both mental health outcomes (depression, anxiety) and suicide risk factors (defeat, entrapment) were significantly mediated by perceived stress and stress appraisals.

## Mediation Model: Mental-Health Outcomes

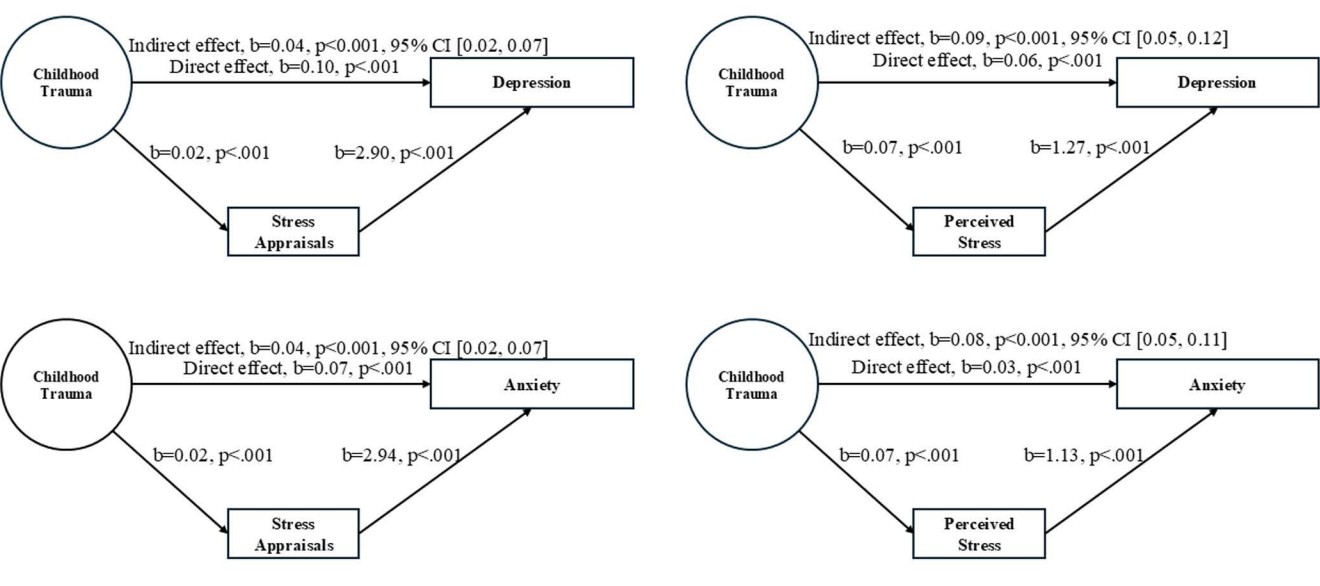

## Mediation Model: Defeat-Entrapment Outcomes

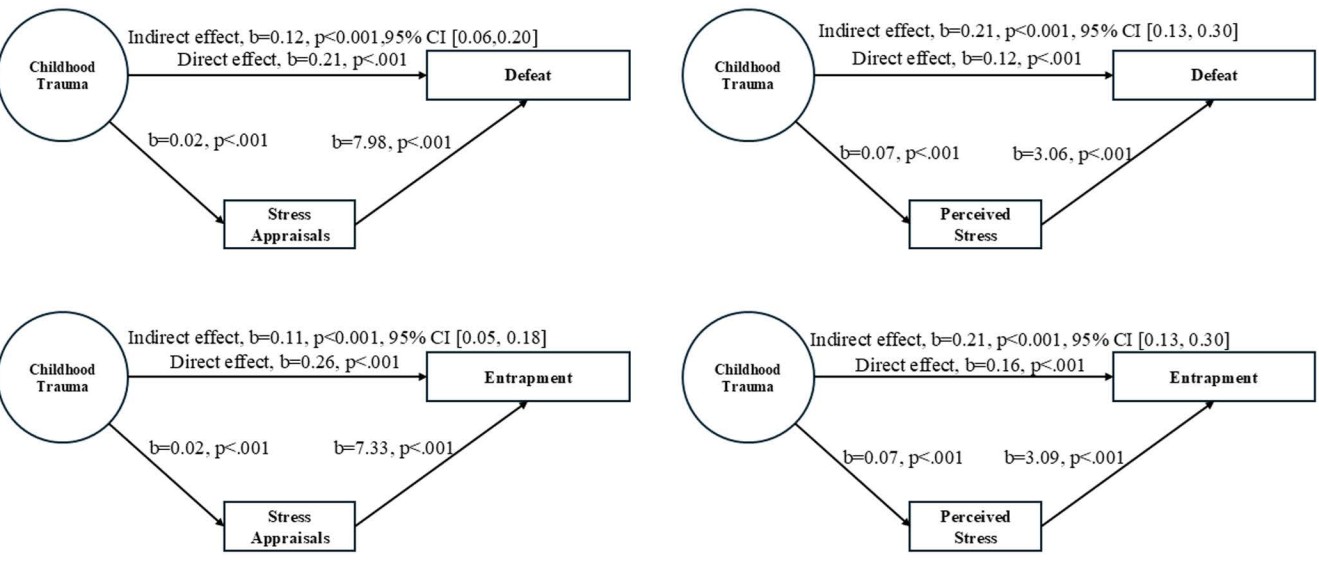

**Fig 1. Path diagrams showing the significant indirect effects of childhood trauma on defeat-entrapment outcomes and mental health outcomes through stress-related variables.** All diagrams represent unstandardised B.

It is important to emphasise that the stress appraisal examined in this study specifically pertains to the evaluation of daily hassles, which are common stressors encountered in everyday life. This appraisal represents a measurement grounded in a highly realistic and ecologically valid context. The higher stress appraisals in response to daily hassles were associated with childhood trauma, but they also point to a potential pathway through which childhood trauma may have a longer lasting and enduring effect on individuals later life. This pathway is also consistent with O'Connor

et al. [19] which found that the effects of childhood trauma on poor sleep quality and greater morning tiredness levels were mediated through higher daily perceived stress. Similarly, Rogerson et al. [9] found that childhood trauma showed significant indirect effects on suicide risk factors (e.g., defeat, entrapment, reason for living) through stress-related variables (e.g., daily stress, sleep quality). Taken together, these findings of indirect pathways are particularly important, because they reveal potential underlying mechanisms by which childhood trauma influences mental health outcomes in later life and increases the risk of suicide. For a long time, one explanation for why childhood trauma exerts such a profound impact on lifelong health was that it disrupts such a crucial period for cognitive, personality, and physiological development, including the emergence of coping and management of stress [13,14]. Findings from the current study extend this understanding by suggesting that cognitive appraisal processes – central to the experience of stress – may also be influenced by childhood trauma, thereby, providing further evidence how childhood trauma may have long term effects on the individual.

## Implications

These current findings may inform interventions designed to reduce the negative effects of childhood trauma. Such interventions should incorporate components targeting stress appraisals which include the perceived stress level, perceived coping level, and the mismatch between them. Furthermore, the treatment strategies for anxiety disorders align with the concept of stressor appraisal measured in this study. Cognitive approaches to anxiety intervention typically focus on two key aspects: i) reducing exaggerated negative perceptions of the outcomes of life events, and ii) enhancing individuals' perceptions of their ability to cope with these life events [58]. The similarity between this core rationale of anxiety intervention and the construct of stress appraisal indirectly underscores the importance of stress appraisal and suggests the feasibility of extending this focus to interventions targeting childhood trauma.

Cognitive-behavioural therapy (CBT) have been shown to have consistent outcomes among interventions for childhood trauma. Among the variants of CBT, the Cognitive Behavioural Analysis System of Psychotherapy (CBASP) has been successfully applied to patients with childhood trauma [59,60], and Cognitive Processing Therapy (CPT) which likewise has been evaluated in childhood trauma survivors [61–63]. Given the core focus of CBT is on challenging maladaptive thoughts and behaviours. It is likely to be particularly suitable for addressing excessive stress appraisals. Therefore, future investigations should aim to design CBT variants that can be delivered by healthcare professionals and incorporate components targeting stress appraisal. These interventions should be also evaluated for their effectiveness in reducing suicide risk among individuals with a history of childhood trauma.

## Limitations

We recognise several limitations in the current research. First, the current study did not screen participants for their current health status, nor did it set inclusion or exclusion criteria on that basis. As a result, the sample likely contained both healthy individuals and patients undergoing treatments, thereby introducing potential confounding factors that may have affected the observed impact of childhood trauma on outcomes.

Other enhancements to the study design could also be beneficial, including the use of a longitudinal study design to track participants' status over time. As outlined earlier, an increasing number of studies are employing daily diary approaches to examine the daily manifestations of stress events on various indicators. Compared to cross-sectional study designs, this ecological methodology yields more robust results. Moreover, although the Daily Stressor Appraisal Scale used in this study demonstrated sound psychometric properties, future research could explore additional methods beyond retrospective scale reports for assessing hassles and include objective assessments of stress (e.g., cortisol levels; [64,65]). This may also include collecting contextual information and conducting thematic analyses, thereby uncovering more detailed insights into how childhood trauma influences stress events in daily life.

Based on the findings of this study, two recommendations are proposed for future research. First, it would be useful to implement more detailed pre-screening criteria for participants, including participants' current and past treatment status (e.g., medication history). Second, future work ought to conduct longitudinal investigations over an extended time frame to examine the effects of daily stress appraisal on a broader range of daily life indicators, thereby providing deeper insights into its long-term impact.

In conclusion, childhood trauma was shown to influence a number of important mental health and suicide risk factors. Our results show that stress appraisal mediates the pathway from childhood trauma to heightened levels of depression and anxiety symptoms, defeat and entrapment. In other words, how a person evaluates stressors contributes to the outcomes of early adversity in adulthood.

## Supporting information

**S1 Fig. Path diagrams showing the non-significant moderation effects on outcome variables through social support, subjective SES, and suicide-related history.** All diagrams represent unstandardised B.
(DOCX)

## Author contributions

**Conceptualization:** Leizhi Wang, Chris Keyworth, Daryl B. O'Connor.

**Data curation:** Leizhi Wang, Chris Keyworth, Daryl B. O'Connor.

**Formal analysis:** Leizhi Wang, Chris Keyworth, Daryl B. O'Connor.

**Investigation:** Leizhi Wang, Chris Keyworth, Daryl B. O'Connor.

**Methodology:** Leizhi Wang, Chris Keyworth, Daryl B. O'Connor.

**Project administration:** Leizhi Wang, Chris Keyworth, Daryl B. O'Connor.

**Supervision:** Chris Keyworth, Daryl B. O'Connor.

**Validation:** Leizhi Wang, Chris Keyworth, Daryl B. O'Connor.

**Visualization:** Leizhi Wang, Chris Keyworth, Daryl B. O'Connor.

**Writing – original draft:** Leizhi Wang, Chris Keyworth, Daryl B. O'Connor.

**Writing – review & editing:** Leizhi Wang, Chris Keyworth, Daryl B. O'Connor.

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
