## [Decision Letter · Decision Letter 0]

16 Apr 2025

Dear Dr. WANG,

Thank you for submitting your manuscript to PLOS ONE. After careful consideration, we feel that it has merit but does not fully meet PLOS ONE’s publication criteria as it currently stands. Therefore, we invite you to submit a revised version of the manuscript that addresses the points raised during the review process.

We look forward to receiving your revised manuscript.

Kind regards,

Marianna Mazza

Academic Editor

PLOS ONE

2. In the online submission form, you indicated that [Any individual or organization seeking access to the data used in this study may obtain it by sending a request to the corresponding author at [ps22lw2@leeds.ac.uk].].

Additional Editor Comments (if provided):

Reviewers' comments:

Reviewer's Responses to Questions

**Comments to the Author**

1. Is the manuscript technically sound, and do the data support the conclusions?

Reviewer #1: Yes

Reviewer #2: Yes

2. Has the statistical analysis been performed appropriately and rigorously?

Reviewer #1: Yes

Reviewer #2: Yes

3. Have the authors made all data underlying the findings in their manuscript fully available?

Reviewer #1: Yes

Reviewer #2: Yes

4. Is the manuscript presented in an intelligible fashion and written in standard English?

Reviewer #1: Yes

Reviewer #2: Yes

Reviewer #1: The manuscript was clearly written and well presented. the analyses was rigorous and adequate and the results was well presented. The study provides robust insight into the impact of childhood trauma on mental outcomes, suicidal risks and stress in later life.

Question for the authors: What informed the choice of one week between session 1 and session 2?

Reviewer #2: Abstract

Lines 33–37

"The current study sought to: (1) investigate whether childhood trauma was associated to mental health outcomes (depression, anxiety), suicide risk factors (defeat, entrapment), and stress related outcomes (stress appraisal, perceived stress) in adulthood and (2) explore the indirect effects of childhood trauma on mental health outcomes and suicide risk factors through stress appraisal and perceived stress."

Clearly defines two aims.

Suggested revision for conciseness and grammar:

"...associated with mental health outcomes..." (Correct "associated to" → "associated with")

Line 38–41

"273 participants were recruited to online prospective study that consist of two sessions with one week interval. In session 1, the childhood trauma questionnaire was completed. In session 2, participants completed measures the level of depression, anxiety, defeat, entrapment, stress appraisal and perceived stress over the preceding week."

Needs grammatical fixes:

"to an online prospective study"

"that consisted of two sessions"

"participants completed measures assessing the level of..."

Introduction

Summary of Strengths

Well-organized and thorough.

Cites foundational and recent literature.

Clearly identifies research gaps.

Introduces relevant theories (IMV model, transactional model).

Logical flow from general to specific.

Areas for Improvement

Line Issue Suggestion

94 Wordiness Reword to improve clarity: "how individuals appraise daily stressors"

101 Diction Replace "quotidian" with "everyday" unless required by journal style

136 Grammar "remain" → "remained"

140–146 Readability Consider using bullet points or shorter sentences to list the study aims

METHOD

Strengths Summary

Clear structure and well-defined procedure.

Ethical and transparent throughout.

Inclusion/exclusion criteria and compensation are well explained.

Pre-registration and REC approval enhance scientific credibility.

Suggestions Summary

Line Issue Suggestion

149 Formality Replace "&" with “and” in subheading.

153 Clarity Consider simplifying "lifetime history of suicide-related behaviours".

Discussion

Overall Strengths

Well-organized and comprehensive.

Strong alignment with theoretical frameworks (e.g., IMV model, transactional stress model).

Integrates prior studies effectively.

Addresses both direct and indirect effects with nuance.

Areas for Improvement

Line Issue Suggested Revision

340 Placement of statistical control variables Move “controlling for age and sex” to methods/results

347 Grammar “Findings from current study are consistent...”

351 Phrasing “associated to” → “associated with”

357 Awkward phrasing “Worthy noticing” → “worth noting”

364 Diction “happened in early life” → “experienced in early life”

375 Typo “stres” → “stress”

386 Grammar “shown” → “showed”

394 Verb form “investigate” → “investigating”

396 Subject-verb agreement “have long term effect” → “has long-term effects”

Implications

Overall Strengths

Effectively links findings to real-world clinical applications.

Clearly grounded in evidence-based therapy, especially CBT.

Highlights mechanisms (stress appraisal) that can be targeted in intervention.

Suggested Revisions Summary

Line Issue Suggested Fix

398 Redundancy “may have implications for potential interventions” → simplify

399–400 Grammar “target at stress appraisals” → “targeting stress appraisals”

403 Awkward phrasing “life events' outcomes” → “outcomes of life events”

405 Vague reference Specify “this” more clearly

410 Citation clarity Integrate citation into sentence more smoothly

413 Grammar “core of CBT on” → “core focus of CBT is on”

Overall Strengths

Clear acknowledgment of sample and design limitations.

Well-structured recommendations for future research.

Highlights mediating mechanisms and theoretical contributions.

Suggestions are realistic, thoughtful, and grounded in current trends (e.g., diary methods, thematic analysis).

Suggested Revisions Summary

Line Issue Suggested Fix

420 Missing article “current sample” → “the current sample”

421 Clarity Rephrase to clarify screening and inclusion

429 Typo Remove duplicate “the”

437 Grammar/clarity “information on undergoing treatment” → “participants’ treatment status”

445 Repetition Combine/simplify stress appraisal discussion

**Do you want your identity to be public for this peer review?** For information about this choice, including consent withdrawal, please see our Privacy Policy

Reviewer #1: **Yes: ** Yetunde Adeniyi

Reviewer #2: **Yes: ** Dr. Kwabena Acheampong

---

## [Author Response · Author response to Decision Letter 1]

20 May 2025

Dear Dr Acheampong,

Thank you for the feedback on the above paper and for the opportunity to submit a revision for further consideration. We have responded to all of the suggestions and we feel that the manuscript is now substantially improved. We hope that you will agree that we have been able to satisfactorily address the comments or provide justification where an alternative approach is taken and that you feel the manuscript is suitable for publication in PLOS ONE. Of course, we are very happy to respond to any further comments. The details of how we have addressed the specific issues raised are below and highlighted in yellow in the manuscript.

PONE-D-25-11717

Reviewer #1: The manuscript was clearly written and well presented. the analyses was rigorous and adequate and the results was well presented. The study provides robust insight into the impact of childhood trauma on mental outcomes, suicidal risks and stress in later life

RESPONSE: We thank the reviewer for this positive comments.

Reviewer #2

Question for the authors: What informed the choice of one week between session 1 and session 2?

RESPONSE: Thanks for your question. We adopted a 1-week prospective design because we wanted to ensure that our assessment of childhood trauma was separated in time from the outcomes variables to help ensure completion of this measure did not contaminate the completion of the other measures as well as allowing us to more robustly test our predictions. One week was chosen specifically because we wanted to reduce the length of time commitment involved in taking part in the study, as our previous experience has indicated that having 7 days between timepoints helps to reduce drop-out.

Abstract

Lines 33–37

"The current study sought to: (1) investigate whether childhood trauma was associated to mental health outcomes (depression, anxiety), suicide risk factors (defeat, entrapment), and stress related outcomes (stress appraisal, perceived stress) in adulthood and (2) explore the indirect effects of childhood trauma on mental health outcomes and suicide risk factors through stress appraisal and perceived stress."

Clearly defines two aims.

Suggested revision for conciseness and grammar:

"...associated with mental health outcomes..." (Correct "associated to" → "associated with")

RESPONSE: As suggested, we have revised the aim to improve conciseness and grammar. Please refer to p. 3.

Line 38–41

"273 participants were recruited to online prospective study that consist of two sessions with one week interval. In session 1, the childhood trauma questionnaire was completed. In session 2, participants completed measures the level of depression, anxiety, defeat, entrapment, stress appraisal and perceived stress over the preceding week."

Needs grammatical fixes:

"to an online prospective study"

"that consisted of two sessions"

"participants completed measures assessing the level of..."

RESPONSE: Thank you very much for pointing this out. We have corrected the grammatical errors mentioned above (see p. 3).

Introduction

Summary of Strengths

Well-organized and thorough.

Cites foundational and recent literature.

Clearly identifies research gaps.

Introduces relevant theories (IMV model, transactional model).

Logical flow from general to specific.

RESPONSE: Thank you for your positive comments.

Areas for Improvement

Line Issue Suggestion

94 Wordiness Reword to improve clarity: "how individuals appraise daily stressors"

101 Diction Replace "quotidian" with "everyday" unless required by journal style

136 Grammar "remain" → "remained"

140–146 Readability Consider using bullet points or shorter sentences to list the study aims

RESPONSE: Thank you very much for your suggestions. We have corrected the grammatical issues you identified, replaced “quotidian” with “everyday” in line 101, and rewritten lines 94 and 140–146 to improve clarity and conciseness. Please see p. 6 & 8.

METHOD

Strengths Summary

Clear structure and well-defined procedure.

Ethical and transparent throughout.

Inclusion/exclusion criteria and compensation are well explained.

Pre-registration and REC approval enhance scientific credibility.

RESPONSE Thank you for your positive comments.

Suggestions Summary

Line Issue Suggestion

149 Formality Replace "&" with “and” in subheading.

153 Clarity Consider simplifying "lifetime history of suicide-related behaviours".

RESPONSE: Thank you very much for highlighting this issue. We have now addressed the problem—please see p. 9.

Discussion

Overall Strengths

Well-organized and comprehensive.

Strong alignment with theoretical frameworks (e.g., IMV model, transactional stress model).

Integrates prior studies effectively.

Addresses both direct and indirect effects with nuance.

RESPONSE Thank you for your positive comments.

Areas for Improvement

Line Issue Suggested Revision

340 Placement of statistical control variables Move “controlling for age and sex” to methods/results

347 Grammar “Findings from current study are consistent...”

351 Phrasing “associated to” → “associated with”

357 Awkward phrasing “Worthy noticing” → “worth noting”

364 Diction “happened in early life” → “experienced in early life”

375 Typo “stres” → “stress”

386 Grammar “shown” → “showed”

394 Verb form “investigate” → “investigating”

396 Subject-verb agreement “have long term effect” → “has long-term effects”

RESPONSE: Thank you very much for highlighting theseareas for improvement. We have implemented all the suggested changes. Please see p. 21-24.

Implications

Overall Strengths

Effectively links findings to real-world clinical applications.

Clearly grounded in evidence-based therapy, especially CBT.

Highlights mechanisms (stress appraisal) that can be targeted in intervention.

RESPONSE Thank you for your positive comments.

Suggested Revisions Summary

Line Issue Suggested Fix

398 Redundancy “may have implications for potential interventions” → simplify

399–400 Grammar “target at stress appraisals” → “targeting stress appraisals”

403 Awkward phrasing “life events' outcomes” → “outcomes of life events”

405 Vague reference Specify “this” more clearly

410 Citation clarity Integrate citation into sentence more smoothly

413 Grammar “core of CBT on” → “core focus of CBT is on”

RESPONSE: Thank you for your helpful suggestions. We have corrected the grammatical errors and awkward phrasing you identified. The section on CBT (lines 412–416) has been rewritten to integrate the citations more smoothly, and the wording at lines 401 and 408 has been revised for greater clarity. Please refer to p. 24 for the detailed changes.

Overall Strengths

Clear acknowledgment of sample and design limitations.

Well-structured recommendations for future research.

Highlights mediating mechanisms and theoretical contributions.

Suggestions are realistic, thoughtful, and grounded in current trends (e.g., diary methods, thematic analysis).

RESPONSE Thank you for your positive comments.

Suggested Revisions Summary

Line Issue Suggested Fix

420 Missing article “current sample” → “the current sample”

421 Clarity Rephrase to clarify screening and inclusion

429 Typo Remove duplicate “the”

437 Grammar/clarity “information on undergoing treatment” → “participants’ treatment status”

445 Repetition Combine/simplify stress appraisal discussion

RESPONSE: Thank you very much for your suggestions. We have corrected the grammatical and spelling errors you noted, including missing articles and typographical mistakes. In addition, we have rephrased the discussion of limitations concerning the screening and inclusion criteria (lines 423–427) to improve clarity. Finally, we have revised the summary of the stress-appraisal findings (lines 444–447) to make it more concise and straightforward. Please see pp. 25–26.

Overall Remarks

This article provides a thoughtful and empirically grounded exploration of the long-term psychological consequences of childhood trauma, with a particular focus on how stress appraisal and perceived stress mediate its effects on mental health and suicide risk factors. The study is timely, well-structured, and contributes meaningfully to the growing body of literature examining the cognitive and affective mechanisms linking early adversity to adult psychopathology.

The use of a two-session prospective design, inclusion of validated psychometric tools, and attention to both direct and indirect pathways of influence (e.g., via perceived stress and stress appraisals) are notable strengths. The study’s grounding in theoretical frameworks such as the Integrated Motivational-Volitional (IMV) model further strengthens its relevance and interpretative depth.

RESPONSE: Thank you so much for your positive comments.

However, several methodological limitations reduce the generalizability of the findings. Notably, the lack of clinical screening for participants introduces the risk of confounding due to varying mental health statuses. Moreover, the cross-sectional nature of the stress appraisal measurement limits causal inferences, though the authors commendably suggest future longitudinal and ecological designs.

RESPONSE: Thank you for your thoughtful comments. As you have noted—and as we detail in the study’s limitations—this research does have certain constraints. These limitations are explicitly acknowledged and discussed in the Discussion section. We believe they do not materially weaken the findings and will instead contribute meaningfully to the existing literature and prompt further research in this area.

The discussion and implications are particularly strong, connecting empirical results to clinical intervention strategies, especially in the context of cognitive-behavioral therapy (CBT). The emphasis on refining CBT to better address maladaptive stress appraisals presents a promising avenue for intervention development.

RESPONSE: Thank you so much for your positive comments.

The writing is generally clear, though some grammatical errors and awkward phrasings (e.g., repetition, article use, and sentence clarity) slightly detract from the professional tone. These could be easily corrected with careful proofreading.

RESPONSE: Thank you very much for all your suggestions on grammar and phrasing. We have incorporated every comment, and have made additional edits throughout to improve the sentence clarify and grammar. We are confident that the revised manuscript is now substantially improved.

---

## [Editor Report · Decision Letter 1]

25 May 2025

Effects of childhood trauma on mental health outcomes, suicide risk factors and stress appraisals in adulthood

PONE-D-25-11717R1

Dear Dr. WANG,

We’re pleased to inform you that your manuscript has been judged scientifically suitable for publication and will be formally accepted for publication once it meets all outstanding technical requirements.

Kind regards,

Marianna Mazza

Academic Editor

PLOS ONE
---

## [Editor Report · Acceptance letter]

PONE-D-25-11717R1

PLOS ONE

Dear Dr. WANG,

I'm pleased to inform you that your manuscript has been deemed suitable for publication in PLOS ONE. Congratulations! Your manuscript is now being handed over to our production team.

Kind regards,

on behalf of

Dr. Marianna Mazza

Academic Editor

PLOS ONE